# BETTER KNOWLEDGE RETENTION THROUGH METRIC LEARNING

## ABSTRACT

In a continual learning setting, new categories may be introduced over time, and an ideal learning system should perform well on both the original categories and the new categories. While deep neural nets have achieved resounding success in the classical setting, they are known to forget about knowledge acquired in prior episodes of learning if the examples encountered in the current episode of learning are drastically different from those encountered in prior episodes. This makes deep neural nets ill-suited to continual learning. In this paper, we propose a new model that can both leverage the expressive power of deep neural nets and is resilient to forgetting when new categories are introduced. We demonstrate an improvement in terms of accuracy on original classes compared to a vanilla deep neural net.

## 1 INTRODUCTION

Despite the multitude of successes of deep neural networks in recent years, one of the major unsolved issues is their difficulty in adapting to new tasks while retaining knowledge acquired from old ones. This problem is what continual learning aims to tackle. To successfully develop a continual learning method, one major limitation that must be overcome is the problem of *catastrophic forgetting* (McCloskey & Cohen, 1989), which describes the phenomenon that a machine learning model finetuned for a new task performs poorly on the old task it was originally trained on.

Various continual learning methods make different assumptions about whether the task identities and boundaries are known at training time and test time (van de Ven & Tolias, 2019; Hsu et al., 2018; Zeno et al., 2018). In the most restrictive setting, task identities are known at both training and test time and the method is only asked to classify among the possible classes *within* the given task, a setting known as *task learning*. In *domain learning*, the assumption that the task identity is known at test time is removed (with all others in place) and each task is assumed to have the same number of classes with similar semantics; in this case, the method is asked to classify among the possible classes within any given task. *Class learning* differs from *domain learning* in that each task may contain a different number of classes that may have non-overlapping semantics with the classes in other tasks, and the method is asked to classify among the possible classes across *all* tasks. *Discrete task-agnostic learning* further generalizes this by removing the assumption of known task identity at training time and replacing it with an assumption of known boundaries between different tasks at training time. In the most general setting, task boundaries between different tasks are also unknown, even at training time. This setting is known as *continuous task-agnostic learning*, which is the setting we focus on in this paper.

Note that the same method can yield very different performance under different settings; in particular, some methods may work very well on more restrictive settings, but undergo a significant performance degradation under less restrictive settings. This is because different underlying strategies employed by various methods may be especially dependent on certain assumptions. As a result, care must be taken when interpreting results across different papers to ensure that evaluation is conducted under the same setting. Refer to (van de Ven & Tolias, 2019; Hsu et al., 2018) for an extensive study and discussion on the performance of various methods under different settings.

Existing continual learning methods can be divided into two broad categories: those that bias the parameters towards parameter values learned on old tasks, and those that expands the model size to accommodate new tasks. In this paper, we propose a new approach that is orthogonal to these categories. Our approach neither penalizes parameter changes nor expands model size. Our key

intuition is that neural nets forget because parameters at all layers need to change to adapt to new tasks. Therefore, one way to improve retention is to keep parameter changes localized. Because upper-layer parameters depend on lower-layer parameters, upper-layer parameters must change when lower-layer parameters change; hence, lower-layer parameters must be kept relatively static to prevent parameter changes throughout the network. One way to achieve this would be to explicitly regularize parameter changes in the lower layers; this is less than ideal, however, because it would reduce the network's expressive power and therefore its capability to learn on new tasks. How do we achieve this without compromising on expressive power?

To arrive at a solution to this problem, we need to consider the underlying cause of parameter changes at *all* layers when fine-tuning on new tasks. In a neural net, each layer on its own has quite limited expressive power and is simply a linear model. As a result, if the features computed by the penultimate layer on training examples for the new task are not linearly separable, then the parameters in the layers up to the penultimate layer must change to successfully learn the new task. To avoid this, we can make the last layer more expressive and replace it with a nonlinear classifier. To this end, we propose replacing the softmax activation layer with a $k$-nearest neighbour classifier. We will show that this simple modification is surprisingly effective at reducing catastrophic forgetting.

## 2 RELATED WORK

Our work offers a novel solution to the continual learning problem by leveraging ideas from metric learning.

**Continual learning** is the idea that machine learning models should be able to emulate humans' ability to learn new information while retaining previously learned skills or knowledge. This capability is generally relevant – for example, in robotics it is desirable for robots to adapt to a given environment while retaining a general functionality across environments; another example from computer vision is the desired ability to update a classifier online with information on new classes while retaining the ability to detect old classes. While it is always possible to accomplish this goal of continual learning through retraining a model on the combined old and new datasets (e.g., as in (Caruana, 1997)), this approach is computationally expensive or infeasible due to the inability to store old data. One commonly used shortcut is to instantiate a model trained on the old dataset and then *finetune* that model on the new data (e.g., as in (Hinton & Salakhutdinov, 2006; Girshick et al., 2014)); however, this method has the side effect of (significantly) decreased performance on the old data.

Recent interest in continual learning has resulted in many proposed methods for continual learning with deep neural networks. They can be grouped into two broad categories: those that dynamically expands the model architecture, and those that constrains the parameter values to be close to those obtained after training on previous tasks.

An example of a method in the former category is (Rusu et al., 2016), which creates a new network for each new task, with connections between the networks (from old task networks to new ones), thus ensuring the network weights for the old network are preserved while enabling adaptation to the new task. Another method (Masse et al., 2018) randomly zeros a subset of hidden units for each task, which can be viewed as a way to implicitly expand the model architecture, since the hidden units that are zeroed are effectively removed from the architecture.

In the latter category are two subtypes of methods: those based on regularization and those based on data replay. Regularization-based methods are all based on the idea of protecting important parameters and introduce different regularizers to penalize changing the values of these parameters from what they were after training on previous tasks. Different methods differ in the way they determine importance of each parameter. Examples of methods of this flavour include elastic weight consolidation (EWC) (Kirkpatrick et al., 2017), online EWC (Schwarz et al., 2018) and synaptic intelligence (SI) (Zenke et al., 2017).

Whereas regularization-based methods can be viewed as constraining the parameter values explicitly, data replay-based methods do so implicitly. Generally speaking, data replay works by generating simulated training data from the model trained on previous tasks and using it to augment the training data used to train on the new task. For example, Learning without Forgetting (LwF) (Li & Hoiem, 2018) performs distillation (Hinton et al., 2015) on the training data for the new task, or in other words, takes simulated labels to be the predictions of the model trained on previous tasks on the

input training data for the new task. Deep Generative Replay (DGR) (Shin et al., 2017) uses a deep generative model to generate simulated input data and labels them with the model trained on previous tasks. Other related methods (Rebuffi et al., 2017; Wu et al., 2018; Venkatesan et al., 2017; van der Ven & Tolias, 2018) use training data from previous tasks, either directly or to train a deep generative model, the samples from which are used to augment the current training set. For a more comprehensive review of this area, we refer readers to (Parisi et al., 2019).

Note that all these methods at least require task boundaries to be known. For example, methods that dynamic expand the model architecture need to know when to expand the architecture, regularization-based methods need to know the point in time at which the parameter values are used to constrain future parameter updates, and data replay-based methods need to know when to generate simulated training data. Some methods, like (Masse et al., 2018), require stronger assumptions, namely that task identities be known as well. In contrast, we consider the setting where neither task identities nor task boundaries are known.

**Metric learning** involves learning a (pseudo-) distance metric. This can be combined with $k$-nearest neighbours ($k$-NN) classifier to yield classification predictions. Standard $k$-NN works by finding the closest $k$ training data points to a test data point and classifying the test data point as the most common label among the training data. The distance metric used for determining the "closest" $k$ points is commonly the Euclidean distance. As proposed by Weinberger et al. (2006), it is possible and desirable to learn a Mahalanobis distance metric that improves the accuracy of $k$-NN. The loss function attempts to enforce clustering of labels with large margins between clusters.

More recent work like (Schroff et al., 2015) has used similar ideas in the context of deep neural networks where the distance between two data points is defined as the Euclidean distance between their embeddings computed by the network. Most methods work by pulling examples whose labels are the same as that of the current training example closer to the training example and pushing examples whose labels are different away. The challenge becomes designing a loss function that is conducive to efficient training of the network. Various loss functions like those used in (Schroff et al., 2015; Tamuz et al., 2011; Van Der Maaten & Weinberger, 2012) have been proposed, to varying degrees of success on varying datasets.

## 3 METHOD

### 3.1 MODEL

The key observation we make is that in order for a neural net to learn on non-linearly separable data, the parameters in the layers below the last layer must change. So, if the training data for the new task is not linearly separable from the training data for the previous tasks, the parameters below the last layer will be changed when training on the new task. Because the parameters in the layer below the last layer are shared across all classes, the classes seen during previous tasks must depend on the parameters below the last layer. And so changes in those parameters would result in a performance drop on these classes, leading to catastrophic forgetting.

To avoid catastrophic forgetting, we must make sure the parameters below the last layer will not change significantly even as the model is trained on new tasks. To this end, we must make the last layer more expressive than a linear classifier. Therefore, we replace it with a nonlinear classifier, namely a $k$-nearest neighbour classifier using a learned Mahalanobis distance metric, which is equivalent to replacing the softmax activation layer in a neural net with a $k$-nearest neighbour classifier using a Euclidean distance metric.

More formally, we can characterize the test-time prediction of a vanilla neural net classifier and the proposed classifier as follows:

$$\Phi_{\text{vanilla}}(x_i) := \arg\max(W_{N \times d} f(x_i) + b_N) \tag{1}$$

$$\Phi_{\text{proposed}}(x_i) := k\text{NN}(W_{D \times d} f(x_i) + b_D), \tag{2}$$

where Eq. 1 is a vanilla feedforward neural network classifier and Eq. 2 is the proposed classifier. Here $x_i$ is the image input, $f(x_i) \in \mathbb{R}^d$ denotes the feature activations of the neural network at the second last layer, $N$ is the number of classes, and $D$ is the dimension of the image embedding output by our model. $\arg\max(\cdot)$ returns the index of the greatest element in the input vector, and $k\text{NN}(\cdot)$

returns the most common label of the $k$ nearest training examples to the input in Euclidean distance. $W_{m \times n}$ denotes a weight matrix of given dimensions $m \times n$ and $b_m$ denotes a bias vector of given dimension $m$.

## 3.2 LEARNING

### 3.2.1 LOSS FUNCTION

Just as the vanilla classifier is trained using cross-entropy loss, to train the proposed classifier, we need to define a loss function. Training using the loss can be viewed as an instantiation of metric learning, where the metric is defined by the parameters of the neural net.

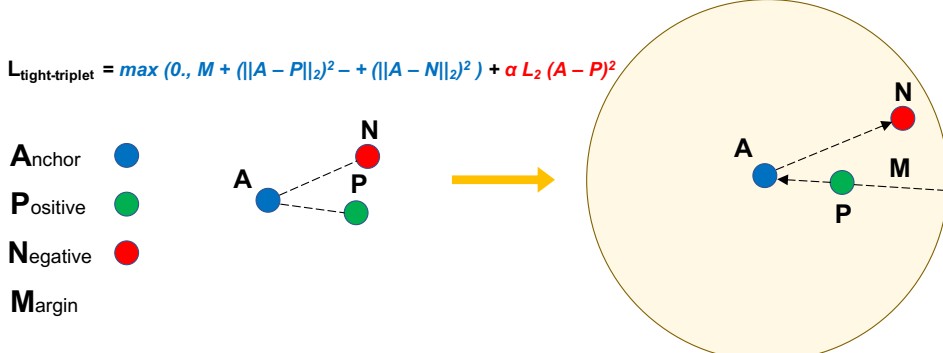

Figure 1: Graphical depiction of the loss function in Eq. 4.

We would like to design the loss so that the predictions of the $k$-nearest neighbour classifier on the feature activations of the last layer are as accurate as possible. To this end, for every training example, we would like the nearby training examples to be in the same class and the training examples that are in other classes to be far away. In practice, we found it suffices to enforce this for only a small subset of training examples, which we will refer to as *anchors*. The training examples that are in the same class as an anchor is known as *positive examples*, and those that are in different classes are known as *negative examples*. In practice, we choose one anchor for each class, which means that we only need to store one example for each class.

The loss function we use is a modified form of the triplet loss (Schroff et al., 2015):

$$L_{\text{triplet}} = \sum_{i=1}^{N} \left[ ||\tilde{\Phi}(x_i^a) - \tilde{\Phi}(x_i^p)||_2^2 - ||\tilde{\Phi}(x_i^a) - \tilde{\Phi}(x_i^n)||_2^2 + M \right]_+. \tag{3}$$

where $\tilde{\Phi}$ denotes the model, i.e.: a function that maps the input to embedding, $x_i^a$ denotes an anchor, $x_i^p$ denotes a positive example and $x_i^n$ denotes a negative example, and $M$ is a hyperparameter representing the desired minimum margin of separation between the positive and negative examples.

Ideally, we would like the training examples of the same class to cluster together. In practice, we found that training using $L_{\text{triplet}}$ results in overlap between clusters for different classes, as shown in Figure 2. To discourage this, we add another term to the loss function to encourage tight clusters:

$$L_{\text{tight-triplet}} = L_{\text{triplet}} + \alpha \sum_{i=1}^{N} ||\tilde{\Phi}(x_i^a) - \tilde{\Phi}(x_i^p)||_2^2, \tag{4}$$

where $\alpha$ is a hyperparameter for balancing different terms in the loss.

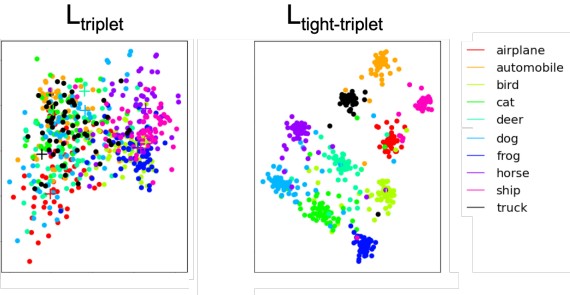

Figure 2: Comparison of the two loss function variants in Eq. 3 and 4. Here, a 5-layer CNN is trained on the CIFAR-10 dataset to output a 2-dimensional embedding. The labeled loss function is used to train the network for each figure. The displayed figures generated by sampling training examples and plotting their embeddings. Plots of test examples appear similar, albeit slightly more spread out for both loss functions.

### 3.2.2 TRAINING PROCEDURE

---
**Algorithm 1** Training Procedure

---
Sample set $X_a$ of $k_a$ anchors
**for** $l = 1, \ldots$ **do**
    Sample subset $X_e$ of $k_e$ training examples
    **for** $x_{ai} \in X_a$ **do**
        $X_p \leftarrow$ set of positive examples of $x_{ai}$ in $X_e$
        $X_n \leftarrow$ set of negative examples of $x_{ai}$ in $X_e$
        $D_{ai,pj} \leftarrow ||\tilde{\Phi}(x_{ai}) - \tilde{\Phi}(x_{pj})||_2^2 \qquad \forall x_{ai} \in X_a, x_{pj} \in X_p$
        $D_{ai,nk} \leftarrow ||\tilde{\Phi}(x_{ai}) - \tilde{\Phi}(x_{nk})||_2^2 \qquad \forall x_{ai} \in X_a, x_{nk} \in X_n$
    **end for**
    **for** $x_{ai} \in X_a$ **do**
        $T_{x_{ai}} \leftarrow \{(x_{ai}, x_{pj}, x_{nk}) : x_{pj} \in X_p, x_{nk} \in X_n, \; D_{ai,pj} < D_{ai,nk}\}$
        $\tilde{T}_{x_{ai}} := \{(t_a, t_p, t_n) \in T_{x_{ai}} : t_n$ is among the $d$ closest negative examples to $t_a$ in $T_{x_{ai}}\}$
        Train model $\tilde{\Phi}$ on $\tilde{T}_{x_{ai}}$ for $m$ steps
    **end for**
**end for**

---

As shown in Figure 1, to train the model, we take a random training example from each class and combine them into a set of anchors. In each iteration, we select a batch of training examples and compute the distances between each anchor and the training examples in the batch. Then we construct the minibatch of triplets used to train the model. We only include triplets where the positive example is closer to the anchor than the negative example and the said negative example is within the $d$ closest to the anchor among negative examples that are farther from the anchor than the positive example. The variable $d$ is a hyperparameter and is known as the *dynamic margin*, because the distance to the farthest negative example to the anchor that is included varies depending on the scale of the embedding. The reason we do not include triplets where the positive example is farther from the anchor than the negative example is because the model would tend to map all examples to zeros otherwise, which is in line with the observations of (Schroff et al., 2015).

While training, we need to retrieve the closest examples to the anchor. To this end, we use an efficient $k$-nearest neighbour search algorithm that is scalable to large datasets and high dimensions known as Prioritized DCI (Li & Malik, 2017).

### 3.3 EMBEDDING NORMALIZATION

We normalize the output embedding of our model, so that the embedding vector for each example has an $\ell_2$ norm of 1. This can be interpreted as making sure $k$-nearest neighbors returns the $k$ closest

| Method | Set A: *Original* | Set A: *After Seeing Set B* (Difference from *Original*) | Set B |
|---|---|---|---|
| Our Method | 97.5% | 95.1% (-2.4%) | 95.6% |
| *Very Forgetful & Adaptive:* | | | |
| SGD with dropout (ratio: 0.5) | 98.62% | 0.0% (-98.62%) | 98.62% |
| Elastic weight consolidation (EWC) | 98.62% | 0.0% (-98.62%) | 98.46% |
| Online EWC | 98.62% | 0.0% (-98.62%) | 98.46% |
| Synaptic intelligence (SI) | 98.62% | 0.0% (-98.62%) | 98.13% |
| *Forgetful & Adaptive:* | | | |
| Deep Generative Replay (DGR) | 98.78% | 89.32% (-9.46%) | 98.40% |
| DGR with distillation | 98.78% | 91.70% (-7.08%) | 98.05% |
| Replay-trough-Feedback (RtF) | 98.96% | 93.22% (-5.74%) | 98.62% |
| *Not Forgetful & Not Adaptive:* | | | |
| Learning without Forgetting (LwF) | 98.62% | 98.07% (-0.55%) | 6.84% |

Table 1: Performance of the proposed method compared to eight other continual learning methods on MNIST. SGD with dropout, EWC, online EWC and SI suffer from catastrophic forgetting on Set A after seeing Set B, but were able to adapt well to the new task (Set B). DGR, DGR + distillation and RtF forget more on Set A compared to our method. LwF retains knowledge learned on Set A; however, it fails to adapt to Set B. We also note that all baseline methods assume knowledge of task boundaries, whereas the proposed method does not.

vectors by *angle*, which in this case is the same as return the vectors that are closest by Euclidean distance. So, our classifier is essentially basing its decision on the maximum cosine similarity between the image embeddings.

## 4 EXPERIMENTS

We consider the setting where training data from a subset of categories are presented to the learner in stages. In the first stage, a subset of the categories are presented (which will henceforth be referred to as Set A), and the learner can iterate over training data from these categories as many times as it likes. In the second stage, the remaining categories are presented (which will henceforth be known as Set B); while the learner can look at training data from these categories as often as it desires, it no longer has access to the training data from the categories presented in the first stage.

### 4.1 TRAINING AND TESTING PROCEDURE

We consider the following training procedure:

1. Train on Set A.
2. Starting from the parameters obtained from the first step, train on Set B.

We evaluate performance on Set A after each step in order to assess the extent each method retains its performance on Set A after training on Set B.

### 4.2 METHODS

For the baseline, we consider a vanilla neural net classifier with softmax activation in the final layer trained with cross-entropy loss. To train on Set B, because the weights of the last layer are specific to categories in Set A, we do not use them and instead initialize the last layer randomly. After training on Set B, for the purposes of evaluation on Set A, we discard the weights of the last layer and train a new set of the weights for the last layer (while keeping the weights of all other layers fixed) on Set A.

For the proposed method, which replaces the softmax activation with a $k$-nearest neighbour classifier, we use the same architecture as the baseline when training on the smaller set (which is our case is always Set B). Thus, the number of parameters is identical to that used by the baseline. To train on

| Method | Set A: *Original* | Set A: *After Seeing Set B* (Difference from *Original*) | Set B |
|---|---|---|---|
| Our Method | 72.3% | 62.3% (-10.0%) | 64.2% |
| *Forgetful & Adaptive:* | | | |
| SGD with dropout (ratio: 0.5) | 64.02% | 0.0% (-64.02%) | 85.05% |
| Elastic weight consolidation (EWC) | 67.38% | 0.0% (-67.38%) | 76.08% |
| Online EWC | 68.97% | 0.0% (-68.97%) | 79.17% |
| Synaptic intelligence (SI) | 68.47% | 0.0% (-68.47%) | 82.83% |
| Deep Generative Replay (DGR) | 69.13% | 0.40% (-68.73%) | 83.75% |
| DGR with distillation | 68.22% | 0.03% (-68.19%) | 85.00% |
| Replay-trough-Feedback (RtF) | 66.57% | 0.20% (-66.37%) | 86.60% |
| *Not Forgetful & Not Adaptive:* | | | |
| Learning without Forgetting (LwF) | 69.28% | 45.50% (-23.78%) | 55.53% |

Table 2: Performance of the proposed method compared to eight other continual learning methods on CIFAR-10. SGD with dropout, EWC, online EWC and SI suffer from catastrophic forgetting on Set A after seeing Set B, but were able to adapt well to the new task (Set B). DGR, DGR + distillation and RtF also forget on Set A, this could relate to the fact that CIFAR-10 data is more complex and it is difficult for these methods to train a generative model for replaying. LwF retains some knowledge learned on Set A; however, it fails to adapt to Set B compared to our method.

Set A, to reduce the amount of training time on large datasets, we first pretrain using cross-entropy loss and then switch to the proposed method later on. To train on Set B, unlike for the baseline, we do not need to do anything special for the last layer of weights, because the output dimension of the last layer does not need to change when the number of classes changes.

On MNIST, which is the dataset most existing continual learning methods are tuned for, we also compare to seven continual learning methods that operate under more restrictive assumptions (where knowledge of task boundaries is required), including Elastic Weight Consolidation (EWC), online EWC, Synaptic Intelligence (SI), Deep Generative Replay (DGR), DGR with Distillation, Replay-through-Feedback (RtF) and Learning without Forgetting (LwF).

## 4.3 DATASETS

We consider four different datasets/splits:

**MNIST**   We first consider MNIST, which is the simplest dataset. We divide it into 2 tasks: Set A consists of the odd digits and Set B includes the even digits.

**CIFAR-10**   We then consider CIFAR-10 and take Set A to be the six animal categories (bird, cat, deer, dog, frog and horse), and set B to be the four remaining categories corresponding to man-made objects (airplane, automobile, ship, truck). This split is designed to be more challenging than just a random split, since we don't expect many intermediate-level features to be shared across Sets A and B, making it easier to forget about Set A when training on Set B.

**CIFAR-10 (5 Tasks)**   We also consider a variant of CIFAR-10 with 5 tasks, where each task corresponds to a pair of consecutive categories. This is more challenging than the two-task setting above, because many more examples would be seen after training on the earlier tasks, and so preventing forgetting on the earlier tasks is harder.

**ImageNet (Random Split)**   Next we consider ImageNet, which is larger and more complex than CIFAR-10. We randomly select 100 classes to serve as Set A, and 100 classes to serve as Set B.

**ImageNet (Dog Split)**   Finally, we consider ImageNet and use a more challenging split. For Set A, we take it to be all 880 categories that are not dogs; for Set B, we take it to be all 120 categories that correspond to particular breeds of dogs. Similar to the split we used for CIFAR-10, this split is designed to be more challenging by minimizing semantic overlap between Sets A and B.

| Method | Task 1 | Task 2 | Task 3 | Task 4 | Task 5 | Average |
|---|---|---|---|---|---|---|
| Our Method | 43.00% | 16.20% | 24.65% | 48.85% | 27.55% | 32.05% |
| *Very Forgetful & Adaptive:* | | | | | | |
| SGD with dropout (ratio: 0.5) | 0.0% | 0.0% | 0.0% | 0.0% | 92.85% | 18.57% |
| Elastic weight consolidation (EWC) | 0.0% | 0.0% | 0.0% | 0.0% | 89.50% | 17.90% |
| Online EWC | 0.0% | 0.0% | 0.0% | 0.55% | 88.30% | 17.77% |
| Synaptic intelligence (SI) | 0.0% | 0.0% | 0.0% | 0.0% | 93.05% | 18.61% |
| *Forgetful & Adaptive:* | | | | | | |
| Deep Generative Replay (DGR) | 0.15% | 1.05% | 0.0% | 0.10% | 93.75% | 19.01% |
| DGR with distillation | 0.05% | 0.70% | 0.0% | 0.15% | 93.15% | 18.81% |
| Replay-trough-Feedback (RtF) | 0.0% | 1.45% | 0.85% | 0.20% | 94.10% | 19.32% |
| *Not Forgetful & Not Adaptive:* | | | | | | |
| Learning without Forgetting (LwF) | 93.55% | 0.65% | 0.0% | 0.0% | 0.0% | 18.84% |

Table 3: Performance of the proposed method compared to eight other continual learning methods on 5-task CIFAR. SGD with dropout, EWC, online EWC and SI suffer from catastrophic forgetting on previous tasks after seeing newer tasks, but were able to adapt well to the last task. DGR, DGR + distillation and RtF forget more on the previous tasks compared to our method. LwF retains knowledge learned on first set; however, it fails to adapt to newer tasks.

| | Our Method | | Baseline | |
|---|---|---|---|---|
| | *Original* | *After Seeing Set B* | *Original* | *After Seeing Set B* |
| ImageNet (Random split) | 69.5% | 59.2% | 72.7% | 30.2% |
| ImageNet (Dog split) | 67.0% | 52.9% | 70.0% | 30.4% |

Table 4: Performance comparison between our method and the vanilla baseline on ImageNet. Performance evaluated as top-1 percent correct on test data for the original task. Baseline represents the network trained with cross-entropy loss. "Original" denotes the network after being trained on the original task (classes in Set A) and "After Seeing Set B" denotes the network after it is fine-tuned on the new task (classes in Set B). See corresponding sections for more details on the dataset and network used for each row.

## 4.4 NETWORK ARCHITECTURE

We used standard network architectures that have proven to work well on the datasets we consider. For MNIST, following (van de Ven & Tolias, 2019), we use a network with two hidden layers of 400 units each. We use this same architecture for our method and all baselines. For CIFAR-10, we use the architecture from the Tensorflow CIFAR-10 tutorial[1], which consists of two convolutional layers followed by three fully-connected layers. For ImageNet, we use the AlexNet architecture (Krizhevsky et al., 2012), which consists of five convolutional layers followed by three fully-connected layers. All networks are trained using Adam (Kingma & Ba, 2014).

## 4.5 KEY FINDINGS

The test accuracy of each method is given in Tables 1 and 4 in the sub-column labelled "Original". After fine-tuning, we compare the accuracy of the network relative to the test-accuracy from set A – these numbers are given in Table 4 in the sub-column labeled "After Seeing Set B". For the MNIST comparison, we also include the test-accuracy from set B.

Table 1 shows the results of the proposed method and existing continual learning methods on MNIST. As shown, regularization-based methods including EWC, online EWC and SI are very forgetful of the knowledge learned on the previous task (set A) but can adapt to the new task (set B). Replay-based methods (DGR, DGR + distillation, RtF) are adaptive as well, but are more forgetful compared to the proposed method. Another replay method, LwF, retains knowledge from the previous task (set A) well, however, it struggles with adapting to the new task (set B). On the other hand, the

---

[1]https://www.tensorflow.org/tutorials/images/deep_cnn

|  | Set A (*Original*) | Set A (*After Seeing set B*) | Set B |
|---|---|---|---|
| Original | 69.5% | 59.2% | 69.2% |
| Number of Positive Examples (5 → 30) | +0.4% | +0.1% | +0.3% |
| Number of Negative Examples (40 → 10) | -0.9% | -1.8% | -0.6% |
| Embedding Dimension (100 → 200) | -0.1% | +0.1% | -0.5% |
| Embedding Dimension (100 → 50) | -1.3% | -0.3% | -0.1% |

Table 5: Test accuracy under the original hyperparameter setting and the change in test accuracy under different hyperparameter settings relative to the original setting. Increasing the number of positive examples (from 5 to 30) slightly improves performance. Decreasing the number of negative examples (from 40 to 10) slightly reduces test accuracy. Increasing the embedding dimension (from 100 to 200) results in similar performance as the original setting, whereas decreasing the embedding dimension (from 100 to 50) moderately lowers the performance. Overall, the method is fairly robust to hyperparameter changes.

proposed method can both adapt to the new task and retain knowledge about the previous task, which is especially remarkable considering its simplicity and ability to operate without knowledge of task boundaries.

Table 4 shows the results of the proposed method and the baseline on more complex datasets, such as CIFAR-10 and ImageNet with random split and dog split. As shown, even though our method achieves a lower test accuracy compared to the baseline after only training on Set A, it is better at *remembering* the knowledge it learned once it is trained on new data. This is true in each of the datasets we tested it on, and the gap is substantial (it achieve nearly double the test accuracy of the baseline) for ImageNet (random split) and CIFAR-10. The gap is smaller on ImageNet (dog split), which can be attributed to the diminished similarity between set A and set B. Since set B only consists of dog images, fine-tuning on this data may wipe out much of the knowledge the model has learned about images in general. In contrast, since ImageNet (random split) does contain a random sampling of images in each set, it is likely that fine-tuning on set B preserves more general concepts about images that are still useful in classifying the images in set A.

## 4.6 ANALYSIS

We further examine the performance of the proposed method on the ImageNet dataset with random split.

### 4.6.1 SENSITIVITY ANALYSIS

We varied the hyperparameter settings and took a look at the impact of changes in hyperparameters. We tried the following settings: reduce number of negative samples, increase number of positive sample and different embedding dimensions. As shown in Table 5, increasing the number of positive examples showed a slight improvement, whereas decreasing the number of negative examples resulted a slight performance drop. A similar trend can be observed with the embedding dimension, where proposed method does not benefit from the increase in embedding dimension, neither does it deteriorate significantly when the embedding dimension is decreased. Overall, we found the method to be reasonably robust to hyperparameter changes.

### 4.6.2 ABLATION STUDY

We tested the proposed method by disabling some key components to determine their contributions. The following components are analyzed: embedding output normalization and dynamic margin. As shown in Table 6, by disabling embedding output normalization, the performance on both tasks drops significantly. When the dynamic margin is removed and replaced with a static margin, the method fails to capture new information from set B. In summary, we found that embedding normalization and dynamic margin contribute enormously to the performance of the proposed method.

|                    | Set A (*Original*) | Set A (*After Seeing set B*) | Set B    |
|--------------------|--------------------|------------------------------|----------|
| With All Components | 69.5%              | 59.2%                        | 69.2%    |
| No Normalization    | -12.2%             | -2.4%                        | -17.8%   |
| No Dynamic Margin   | -9.5%              | +0.8%                        | -14.8%   |

Table 6: Test accuracy with all components, and changes in test accuracy relative to the original proposed method after various components are removed. Not normalizing the embedding output dramatically lowers test accuracy, and especially limits the ability to adapt to the new task (Set B). Disabling dynamic margin also greatly decreases the performance on Set B. Overall, we found that embedding normalization and dynamic margin are crucial for the method to be able to adapt well to new tasks.

## 5 CONCLUSION AND FUTURE WORK

As discussed in Section 4, we achieve consistent results across four different datasets/splits. These results suggest that the method is successful at addressing the catastrophic forgetting problem, and hint at a possible general strategy for reducing catastrophic forgetting, namely the idea of replacing the last layer of a neural net with a non-linear classifier.

In the future, we would like to explore combining the proposed approach with other orthogonal approaches for tackling catastrophic forgetting. We also plan to explore applications that can benefit from continual learning approaches, such as reinforcement learning.

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
