# OpenReview forum: "Better Knowledge Retention through Metric Learning"
_ICLR.cc/2020/Conference — Reject_

### Official Review · AnonReviewer1 · 2019-10-19
**Official Blind Review #1**

**Rating:** 3

**Review:**

This paper considers the use of a metric learning approach in a continual/lifelong classification settings. Experiments show in the case of two tasks forgetting can be minimized by using the approach.

Methods
The proposed method appears to be a standard triplet loss. The authors add a second term to the triplet loss that is essentially making the loss a combination of the triplet and siamese loss. It’s not really explained anywhere why they do this and whether its essential to the performance.

Is there anything specific to continual learning done or is the paper essentially pointing out this existing method (metric learning + nearest neighbor) is surprisingly effective for forgetting. If this is the case the authors should present it in this way I think.

Although triplet loss can often yield reasonably performance on classification problems it tends to not perform as well as cross entropy loss, this is observed in other works as well as this one.

A major question of mine: it is not clear from the method nor experiments what samples are stored after task A for the kNN classifier. Is it all of the data samples from the previous task?

Experiments
The experimental results consider a custom continual learning setup where there is two sets of categories. Overall the experiments seem lacking at the moment in rigorous comparisons.

MNIST experimental comparisons are currently suspect. It is  very surprising that LwF does so poorly, do the authors have some explanation for this. LwF is typically a reasonable baseline for these 2 task settings (e.g. https://arxiv.org/pdf/1704.01920.pdf).  Similarly the well known EWC is shown to simply not work at all for the very task it was designed for on the MNIST dataset. LwF and EWC simply not working to any degree seem to me like  rather dramatic claims to make without any explanation.
Cryptically the fine-tuning baseline described in 4.2 is not shown here for MNIST? This seems a major oversight

CIFAR10/Imagenet Experiments
It is not clear if the baseline finetuning is done on only the top weights or the entire network. Both of these baselines should be considered. Another good baseline to consider is finetuning with cosine distance and only the top weights as in https://arxiv.org/pdf/1804.09458.pdf and other recent works should also be considered

Why do the authors not include any of the baselines from MNIST experiments here, for example LwF.

Ablations study the need for normalization and dynamic margin, it seems these are helpful for accuracy and forward transfer (and not as critical for minimizing forgetting).


The author state their method is agnostic to the task boundaries, its a bit unclear what this means in this context. The procedure is not online and the labels of the samples are being used? If the authors are referring to the need to add additional outputs to the “vanilla” model this seems like it can be trivially addressed by simply saying outputs are added the first time a new class is seen thereby making it agnostic to the boundary in the same sense as this method.

Clarity
Can be problematic at times. Although all the elements of the approach are outlined the motivations are overly wordy and repetitive making them actually hard to follow.

-(minor) first/2nd paragraph of 3.1 seems a bit redundant making it hard to follow

Overall I think the idea to consider metric learning and local adaptation for continual learning is interesting, however the current work is currently lacking in both experimental evidence (appropriate comparisons) and clear motivation/difference to existing work  for its particular instantiation of this idea.

++++Post Rebuttal++++

Thank you for your detailed responses.

The clarification about “task-agnostic” for the experiments does make them look more relevant than I had previously assessed. I do want to note that the language used for this is inconsistent with the ones used in other papers, which typically calls this a “shared-head” setting (https://arxiv.org/pdf/1801.10112.pdf, https://arxiv.org/pdf/1805.09733.pdf, https://arxiv.org/pdf/1903.08671.pdf ). It is also somewhat inconsistent with the authors own definition of “task agnostic learning” given in the introduction of this paper which implies it is something related to task boundaries at training time, in fact this is something related to availability of the task id at test time. I suggest the authors to make this more clear. Furthermore, the authors should highlight all this in the experiment text, e.g. noting EWC does poorly but this is because we use a different protocol than this and this paper etc.

Regarding the experiments under this light they do look more reasonable. Indeed it has been observed that EWC works poorly in the shared-head setting https://arxiv.org/pdf/1801.10112.pdf
Regarding the new 5 task CIFAR-10 the results are interesting, however I will point the authors to the work above (Rwalk) which also reports results in this setting better than theirs (but not by too much).

I do however still have issues regarding the memory usage of the method, specifically which data needs to be stored from previous tasks. It is still not completely clear and I find obfuscated since just one sentence not even fully answering the concern about this was added to the manuscript despite myself and another reviewer asking about it. My understanding based on the (somewhat conflicted responses) of the authors is they store a substantial amount of prior task data, but most of this is only used  at test time. For example for imagenet as much as 1000 images/class are stored for testing time. This begs the question why not use this data for training as well if it is allowed to be used by the model at testing time (and therefore preserved from the first task), why is the storage cost of this data not considered and how do the authors justify this still being a lifelong learning setup. As an alternative, why can't one use a much bigger fully parametric model that uses the same amount of storage as the authors model + stored images. It seems it is not fair to compare these to methods that cant utilize this large storage amount.

Finally its not clear if this data is stored as raw images or somehow stored as embeddings. If it is stored as embeddings this would require some discussion on how the authors avoid representation drift when the next task is training. If the authors store raw images, it means at evaluation time the entire raw dataset needs to be re-encoded, therefore the model can’t perform easily anytime inference.

Unfortunately the discussion period ended but I would have liked more clarification on this, on the other hand these pieces of information should really have been in the manuscript in the first place.

Overall, my impression of the paper is improved.  But I do think it could use some further writing revisions to emphasize/clarify key points: a) the method is not new (it says e.g. in abstract “new model” which is misleading) but its application in CL is under-explored b) the experiments show poor performance on existing methods because most of those are not designed nor work well for the shared head “task agnostic” setting, while metric learning handles it gracefully.  c) be explicit about what is the memory being stored when moving onto the next task (this should be somewhere visible and explicit) and how this is justified



**Experience Assessment:**

I have published one or two papers in this area.

**Review Assessment: Checking Correctness Of Derivations And Theory:**

N/A

**Review Assessment: Checking Correctness Of Experiments:**

I carefully checked the experiments.

**Review Assessment: Thoroughness In Paper Reading:**

I read the paper at least twice and used my best judgement in assessing the paper.

---

> ### Author Response · Authors · 2019-11-15
> **Response to Review 1**
>
> Thanks for your review. Below is our response:
>
> Q1: “The authors add a second term to the triplet loss that is essentially making the loss a combination of the triplet and siamese loss. It’s not really explained anywhere why they do this”
> A1: This is in fact explained in the paper. As mentioned at the bottom of page 4, “In practice, we found that training using L_{triplet} results in overlap between clusters for different classes, as shown in Figure 2. To discourage this, we add another term to the loss function to encourage tight clusters”.
>
> Q2: “Is the paper essentially pointing out this existing method (metric learning + nearest neighbor) is surprisingly effective for forgetting. If this is the case the authors should present it in this way I think.”
> A2: As we stated at the end of the introduction section, “We will show
> that this simple modification is surprisingly effective at reducing catastrophic forgetting”, which is already in line with this suggestion.
>
> The simplicity of our method is an important advantage compared to other methods that add specialized regularizers, because (1) a simpler method is more broadly applicable because it can be used in settings when the signals required by the regularizers (e.g.: task identity or boundary - more on this later) are unavailable, (2) and is more flexible because it can be combined with other approaches.
>
> The insight that metric learning can be used effectively for continual learning is novel and did not appear in prior work to our knowledge - this represents a new perspective for continual learning research that catastrophic forgetting may be partly caused by limitations in the model itself rather than problems with the training objective.
>
> Q3: “Although triplet loss can often yield reasonably performance on classification problems it tends to not perform as well as cross entropy loss, this is observed in other works as well as this one.”
> A3: While this may be true if the goal is to maximize single-task performance, the point of this paper is to demonstrate that metric learning is quite effective if the goal is to minimize catastrophic forgetting. Future improvements to metric learning techniques could help narrow the gap between triplet loss and cross-entropy loss on single tasks, but are orthogonal to our method.
>
> Q4: “A major question of mine: it is not clear from the method nor experiments what samples are stored after task A for the kNN classifier. Is it all of the data samples from the previous task?”
> A4: We only store one example from each class (to serve as anchors).
>
> Q5: “MNIST experimental comparisons are currently suspect. It is  very surprising that LwF does so poorly, do the authors have some explanation for this. LwF is typically a reasonable baseline for these 2 task settings (e.g. https://arxiv.org/pdf/1704.01920.pdf).”
> A5: LwF in the paper the reviewer referenced is evaluated under a different setting than the setting considered in our paper. In the referenced paper, the method is assumed to know which dataset (a.k.a. task) each test example belongs to (i.e.: the task learning setting). As a result, it only needs to discriminate among the classes within that dataset. In our paper, the method is not assumed to know which dataset each test example belongs to (i.e.: the task-agnostic learning setting), and so is required to discriminate among the classes across all datasets. As shown in various prior papers (e.g.: https://arxiv.org/pdf/1810.12488.pdf and https://arxiv.org/pdf/1904.07734.pdf), LwF only achieves an average accuracy in the 20% range under the evaluation setting we consider, whereas it achieves an a near-perfect accuracy in the easier evaluation setting (i.e.: the task learning setting) considered in the original LwF paper.
>
> (continued below...)

---

> > ### Author Response · Authors · 2019-11-15
> > **Response to Review 1 (Continued)**
> >
> >
> > Q6: “Similarly the well known EWC is shown to simply not work at all for the very task it was designed for on the MNIST dataset. LwF and EWC simply not working to any degree seem to me like  rather dramatic claims to make without any explanation.”
> > A6: EWC was again evaluated under a different setting than the setting considered in our paper. In the EWC paper, all the different datasets (a.k.a. tasks) are assumed to share the same classes (i.e.: the domain learning setting), and so the method only needs to discriminate among these classes. On the other hand, in our paper, the classes in each dataset are disjoint (i.e.: the task-agnostic learning setting), and so the method needs to discriminate among all the different classes across datasets. Moreover, EWC was evaluated on *permuted* MNIST in the original paper, whereas it was evaluated on *split* MNIST in our paper. In permuted MNIST, each task is a different random permutation of the pixels of images in MNIST, and all ten classes are represented in each task. In split MNIST, each task is a different subset of MNIST classes. The latter is more challenging because the method does not have access to all ten classes at any given time. See appendix B of https://arxiv.org/pdf/1805.09733.pdf for an explanation of why a method that works well on permuted MNIST may fail on split MNIST. As shown in prior literature (e.g.: https://arxiv.org/pdf/1904.07734.pdf), EWC achieves 94% average accuracy in the domain learning setting (which is consistent with the results in the original EWC paper), but only achieves 64% average accuracy when the dataset is changed to split MNIST. In the harder setting of class learning (which is still easier than the task-agnostic learning setting that we consider, but shares the same evaluation protocol as our setting), the average accuracy of EWC drops to 20% (this suggests 100% accuracy on the most recent task and close to 0% accuracy on four earlier tasks, which is consistent with the results reported in our paper).
> >
> > Q7: “It is not clear if the baseline finetuning is done on only the top weights or the entire network. ”
> > A7: The baseline finetuning was only performed on the top weights (all the convolutional layers are fixed). Finetuning on the entire network leads to even more forgetting.
> >
> > Q8: “Another good baseline to consider is finetuning with cosine distance and only the top weights”
> > A8: As mentioned in Sect. 3.3, normalizing the output embedding vectors and minimizing the Euclidean distance (both of which we do already) is equivalent to maximizing cosine similarity. This is because || u - v ||^2 = <u - v, u - v> = ||u||^2 - 2 <u, v> + ||v||^2 = 2 - 2 <u, v>, and so cosine similarity is a monotonic transformation of Euclidean distance.
> >
> > Q9: “The author state their method is agnostic to the task boundaries, its a bit unclear what this means in this context.”
> > A9: Regularization-based methods like EWC and SI need to estimate how important each parameter is to the previous tasks, which will be used to penalize changes to parameters when training on future tasks. If there weren’t task boundaries, i.e.: different tasks were interlaced, then it is unclear when this estimation should be done. Similarly, methods like LwF need to generate pseudo-labels on the data from the new task from the model trained on previous tasks. If different tasks were interlaced, it is unclear when the pseudo-labels should be generated. This is why methods like EWC, SI and LwF are not applicable to the setting without task boundaries.
> >
> > In our case, the proposed method doesn’t need to do anything when one task ends and another one begins, and so the method can be naturally used in the setting without task boundaries.

---

> > ### Comment · AnonReviewer1 · 2019-11-15
> > **A few quick questions**
> >
> > Thanks for your response I will definitely take it into account and update my review later. I just had a few quick questions:
> >
> > -You state there is only one sample used per class to serve as anchor, but this is also the sample used for the nearest neighbor search at test time? During test time how many samples are being compared (is it one per class?) it seems like the seleciton of these samples would be kind of important.
> >
> > -I was also wondering if you have some thoughts on the relationship to this (admittedly somewhat recent) paper https://arxiv.org/abs/1905.09447

---

> > > ### Author Response · Authors · 2019-11-15
> > > **Thank you for the reply, here is the response:**
> > >
> > > Q1: “You state there is only one sample used per class to serve as anchor, but this is also the sample used for the nearest neighbor search at test time? During test time how many samples are being compared (is it one per class?) “
> > > A1: Yes, those anchors are included during testing. At test time, we use a subset of the training set (e.g. for ImageNet, we used 50 samples per class with k=15, which were not used during training).
> > >
> > > Q2: “relationship to this (admittedly somewhat recent) paper https://arxiv.org/abs/1905.09447 ”
> > > A2: Though both methods adopt metric learning, they have some major differences:
> > > 	1. The referenced paper *trains* on stored training examples from the old tasks when training on a new task, whereas our method does not.
> > > 	2. Our kNN classifier generates the output by finding the most common label of the k nearest training samples, whereas the other method classifies using a softmax over distance in the embedding space.
> > > 	3. Our experiment setting (split MNIST/CIFAR/ImageNet) is more challenging and general compared to the setting in that paper (which they call the Split CIFAR10 “incremental class” setting), where the every subsequent task includes all classes that are in all previous tasks.

---

### Official Review · AnonReviewer3 · 2019-10-22
**Official Blind Review #3**

**Rating:** 3

**Review:**

This paper applies metric learning to reduce catastrophic forgetting on neural networks. By improving the expressiveness of the final layer, the authors claim that lower layers do not change weights as much, leading to better results in continual learning. They provide large-scale experiments on different datasets.

I like the idea that the authors propose and the intuition for why it works, and the paper is well-written. However, I have some concerns and questions. My main concern is that experiments are only performed in the two-task setting, which is highly restrictive.

The authors claim that they tackle the general 'continuous task-agnostic learning' setting. However, they only test on the two-task setting. There are various problems with considering only a two-task setting (see for example Farquhar and Gal, "Towards Robust Evaluations of Continual Learning"). It is too easy to optimise parameters and methods to work in the two-task setting that will not generalise to more than two tasks, which the authors seem to claim. I would need to see experiments on more than two tasks. Aside from this, the experiments seem detailed, with a reasonable baseline, large-scale experiments (on ImageNet), and with an ablation study.

It seems to me like the anchors need to be chosen before training. This means that this method requires memory / storage of past data examples. It is usually fine to do store a small subset of examples in continual learning, but should be made explicit, because it may not always be possible (eg if there are data privacy laws).

I do not understand the reason why the output embeddings need to be normalised (Section 3.3)? I can see from Table 4 that it improves results, but do not see any intuition.

I would also like to see the computational cost of this method, perhaps as a run-time compared to the baseline. There are many hyperparameters to tune on the validation set which may slow the method down. The sensitivity analysis did not consider changing 'd' or 'M', which seem like crucial hyperparameters to me.

------------
EDIT: I will keep my score after the the discussion with authors. Although the paper has improved in my opinion, I still recommend Weak Reject. I very much appreciated the 5-task CIFAR-10 results. However, there are simple baselines in this setting that I believe need to be explored and reported. Namely, baselines but with samples, eg EWC+samples, akin to the RWalk paper that AnonReviewer1 mentions (https://arxiv.org/pdf/1801.10112.pdf). This is because the proposed method also uses samples. Going from the RWalk paper, this improves results for the baselines considerably, but this may depend on number of samples etc. I understand there was not much time during the rebuttal period to include this. I hope that the authors will consider doing so in the future.

The discussion/explanation regarding 'task-agnostic' (train and test time) and also regarding how the anchors are chosen needs to be made clearer.

**Experience Assessment:**

I have published one or two papers in this area.

**Review Assessment: Checking Correctness Of Derivations And Theory:**

I assessed the sensibility of the derivations and theory.

**Review Assessment: Checking Correctness Of Experiments:**

I assessed the sensibility of the experiments.

**Review Assessment: Thoroughness In Paper Reading:**

I read the paper thoroughly.

---

> ### Author Response · Authors · 2019-11-15
> **Response to Review 3**
>
> Thanks for your review. Below is our response:
>
> Q1: Experiments on more than two tasks
> A1: We added new results on a five-task CIFAR-10 dataset (where each task is a consecutive pair of classes), along with results using the baselines, which are presented in Table 2. As shown, our method outperforms all baselines.
>
> Q2: Anchors and requirements for storage
> A2: We only used one anchor per class, so only a minimal number of past data examples need to be stored (we’ve made this clearer in the manuscript). The anchor for a class can be chosen the first time an example from that class is encountered.
>
> Q3: Why normalization helps intuitively
> A3: By normalizing vectors, we essentially project them onto a unit sphere. The benefit of this is that the sphere is a closed surface, unlike the original Euclidean space the vectors lie in. In Euclidean space, all points can easily be pushed very far away from each other, or brought very close together - neither of these scenarios help with classifying points correctly. On the other hand, pushing points away from a point on a unit sphere must make them closer to a point on the opposite side of the sphere - this property of the sphere helps us avoid either of the two scenarios above.
>
> Q4: Computational cost of the method
> A4: Our method ~2 minutes on MNIST and ~12 minutes on CIFAR-10, which are comparable to the runtimes of the baselines.

---

### Official Review · AnonReviewer2 · 2019-10-23
**Official Blind Review #2**

**Rating:** 6

**Review:**

This paper presents a possible way to mitigate catastrophic forgetting by using a k-nearest neighbor (kNN) classifier as the last layer of a neural network as opposed to a SoftMax classifier.  I think this an interesting and possibly novel use of a kNN layer (I haven't seen similar uses although I'm not that familiar with the specific research area).  At the same time it's not presenting a ground breaking new algorithm or anything like that.

Overall the paper is fairly well written and not too hard to follow.  I would say overall results in Table 1 are positive although the authors' approach has the lowest performance after just training on set A if that initial accuracy is important, and also doesn't have quite as high of an accuracy on test B compared to most of the other baselines.  Additionally, if you add the accuracy on both set A and set B after training on set B the sum is slightly higher for Rtf.  If you look at the minimum accuracy between set A and set B after training on set B, however, the authors' method has the highest value which might be what someone is looking to maximize.

One weakness of this is paper is that I think there are other baselines that should be compared against in Table 1 such as something as basic as SGD with dropout (some of the baselines that are compared against in Table 1 were compared against SGD with dropout in their citations).  There are a number of additional approaches outlined in https://www.cs.uic.edu/~liub/lifelong-learning/continual-learning.pdf.  Also maybe even something with self attention such as Serra at al. https://arxiv.org/pdf/1801.01423.pdf.

Another potential issue I have with this paper is that it only reports results for the authors' method and the vanilla baseline for more complex CIFAR-10 and ImageNet data sets in Table 2.  Assuming there aren't restrictive assumptions for some of the methods that prevent them from being run on the other data sets (at least SI was previously evaluated on CIFAR-10), I would like to see how other baselines perform on these more complex datasets too.

The lack of some more baselines such as SGD with dropout, and not reporting the performance of the same baselines from Table 1 in Table 2, cause me to be very borderline on this paper.  I do appreciate the sensitivity analysis and ablation study provided.

As alluded to in future work I'm curious how the authors' approach might be applied to reinforcement learning, and if there could be a way to deal with continuous action spaces in RL.

**Experience Assessment:**

I have read many papers in this area.

**Review Assessment: Checking Correctness Of Derivations And Theory:**

N/A

**Review Assessment: Checking Correctness Of Experiments:**

I assessed the sensibility of the experiments.

**Review Assessment: Thoroughness In Paper Reading:**

I read the paper at least twice and used my best judgement in assessing the paper.

---

> ### Author Response · Authors · 2019-11-15
> **Response to Review 2**
>
> Thanks for your review. Below is our response:
>
> Q1: Interpretation of results in Table 1
> A1: Our method achieves the second highest absolute performance and also the second smallest drop in performance on Set A after seeing Set B. The only method that achieved better performance on Set A after seeing Set B is LwF, which is because its performance on Set B is very poor. This indicates that our method achieves a good balance in learning well on new data and retaining performance on old data. It is also a lot less complex than methods that can achieve reasonable retention performance (DGR, DGR + distillation and RtF) - these methods require training a generative model (a VAE) to produce pseudo-training examples, whereas our method does not require training an external model. This also makes our method more broadly applicable, since it is more difficult to train high-performing generative models on some datasets, e.g.: CIFAR-10 (more on this below).
>
> Q2: Additional SGD + dropout baseline
> A3: We have added this baseline (with a dropout probability of 0.5, which is standard in literature and recommended by https://arxiv.org/abs/1312.6211) in Table 1. As shown, our method outperforms the baseline in terms of retention ability by a large margin.
>
> Q3: Other baselines for CIFAR-10
> A3: We have added results of other baselines on CIFAR-10 in Table 3. As shown, our method achieves the highest absolute performance on Set A after seeing Set B and also the smallest drop in performance on Set A after seeing Set B. Interestingly, on this more complex dataset, none of the baselines (except for LwF) are able to retain significant amounts of knowledge after training on Set B. As discussed above, methods that rely on generative models (DGR, DGR + distillation and RtF) no longer perform well because training high-performing generative models on CIFAR-10 is more difficult due to the increased complexity of the data.
>
> Q4: Extension to the RL setting with continuous action space
> R4: We plan to explore this in future work and consider replacing the k-nearest neighbour classifier with k-nearest neighbour regression. This requires changing the triplet loss to encourage samples within the neighbourhood that have similar outputs as the ground truth to be moved closer, and samples within the neighbourhood that have dissimilar outputs as the ground truth to be moved farther.

---

### Decision · Program_Chairs · 2019-12-19

**Decision:**

Reject

**Comment:**

Catastrophic forgetting in neural networks is a real problem, and this paper suggests a mechanism for avoiding this using a k-nearest neighbor mechanism in the final layer. The reason is that the layers below the last layer should not change significantly when very different data is introduced.

While the idea is interesting none of the reviewers is entirely convinced about the execution and empirical tests, which had partially inconclusive. The reviewers had a number of questions, which were only partially satisfactorily answered. While some of the reviewers had less familiarity with the specific research topic, the seemingly most knowledgeable reviewer does not think the paper is ready for publication.

On balance, I think the paper cannot be accepted in its current state. The idea is interesting, but needs more work.